# Mitigating Forgetting in Online Continual Learning via Instance-Aware Parameterization

**Hung-Jen Chen[1], An-Chieh Cheng[1], Da-Cheng Juan[2], Wei Wei[2], Min Sun[134]**
[1]National Tsing-Hua University, Hsinchu, Taiwan
[2]Google Research, Mountain View, USA , [3]Appier Inc., Taiwan
[4]MOST Joint Research Center for AI Technology and All Vista Healthcare, Taiwan
andyqmongo@gapp.nthu.edu.tw, accheng.tw@gmail.com
{dacheng,wewei}@google.com, sunmin@ee.nthu.edu.tw

## Abstract

*Online continual learning* is a challenging scenario where a model needs to learn from a continuous stream of data without revisiting any previously encountered data instances. The phenomenon of catastrophic forgetting is worsened since the model should not only address the forgetting at the task-level but also at the data instance-level within the same task. To mitigate this, we leverage the concept of "instance awareness" in the neural network, where each data instance is classified by a path in the network searched by the controller from a meta-graph. To preserve the knowledge we learn from previous instances, we proposed a method to protect the path by restricting the gradient updates of one instance from overriding past updates calculated from previous instances if these instances are not similar. On the other hand, it also encourages fine-tuning the path if the incoming instance shares the similarity with previous instances. The mechanism of selecting paths according to instances similarity is naturally determined by the controller, which is compact and online updated. Experimental results show that the proposed method outperforms state-of-the-arts in online continual learning. Furthermore, the proposed method is evaluated against a realistic setting where the boundaries between tasks are blurred. Experimental results confirm that the proposed method outperforms the state-of-the-arts on CIFAR-10, CIFAR-100, and Tiny-ImageNet.

## 1 Introduction

Continual learning aims at finding the method that will prevent a learning model from forgetting previously learned tasks when learning the new task, *i.e.* to overcome the catastrophic forgetting. From the human perspective, we expect that one may have a better understanding if the task is related to previously learned knowledge. For deep learning models, however, knowledge learned from a previous task is often "washed out" by fine-tuning on a different one, making neural networks to exhibit the behavior of catastrophic forgetting. Mitigating forgetting and transferring knowledge between tasks are the key components of continual learning. Although there are plenty of experiment settings proposed by previous works, recent studies such as [1–4] start to solve the more challenging scenario: *online continual learning*. Instead of using the entire training set repeatedly within a task, which is found in the conventional continual learning setting, online continual learning further restricts that each sample can only be seen once. This setting may emulate scenarios when the learner cannot get a sufficient amount of data for tasks at the beginning of the training phase or it is not possible to store the data for the entire task due to privacy issues or memory limitations.

Currently, there are three types of continual learning methods based on how they store the information of previous task: regularization-based, replay-based, and architectural-based. Regularization

approaches such as [5] regularize the network parameter during training to prevent parameters move away from the old tasks. Replay-based approaches replay the previous data stored in a fixed-sized buffer when training on new tasks. Architectural, expansion, parameter-isolation approaches can accommodate the knowledge to prevent new data interference with previous knowledge. We are interested in architectural approaches in our work that can truly satisfy the online learning assumption, i.e., each sample can only be seen once, which is not true for replay-based methods.

One of the reasons for catastrophic forgetting has to do with the fact that updates on a latter task usually wash out the knowledge learned in a previous task. This is a natural consequence of using gradient descent based methods that optimizes directly on an objective defined by a dataset. Models from previous tasks are seen as initialization and will be "washed out" by new updates. This will be more severe for online continual learning, since previous instances in the same task may also be "washed out". We include a baseline, Independent, in the Section 4 to elaborate the idea. One direct way of solving catastrophic forgetting in online continual learning is to learn each instance individually. Nevertheless, it is often difficult to achieve that, due to the lack of amount of data and the unrestricted growing size of the neural network. Also, it is believed that to solve the continual learning problem, the network should have the ability to both reduce the interference and maximal the potential knowledge transfer by weight sharing [6].

We propose a new direction for solving online continual learning with the concept of instance-awareness. For each sample in the neural networks, the controller will assign the most suitable path based on its visual feature. Paths assigned by the controller will receive the images with similar visual features, which is beneficial to help reduce the probability to overwrite the knowledge and improve the performance. Instance-aware has been used in image classification [7–9] for solving the diverse and various samples in the dataset. It is believed that for each given sample, there exists the best model to classify it correctly and efficiently. Inspired by InstaNAS [7], the high-level illustration of the model with instance-awareness is to first prepare a meta-graph, then tailor a child model from the meta-graph for each input sample. In our work, we leverage the instance-aware concept to alleviate catastrophic forgetting for each instance. When training the neural network, the controller will find the best path in the meta-graph for the given sample, then only train the given path and mitigate the forgetting by minimizing the probability to overwrite the existing knowledge from other unrelated paths. By only updating the path with similar visual patterns, our model can both reduce the interference and transfer the existing knowledge to the new instance. Note that the mechanism of selecting paths according to instances similarity is naturally determined by the controller, which is compact and online updated.

The overall concept is illustrated as Figure 1. Here we see two samples in the Figure with two images of butterflies that are similar to each other and an image of a dog that is very different from the other two. We would want the parameters to be updated on the first two images to be similar to each other but not the third one. To achieve that, we store a large meta-graph that consists of $L \times K$ model components, representing a possible universe of architectural spaces that has $L$ layers and $K$ blocks for each layer. We then employ a controller network that dynamically chose the neural network architecture in the meta-graph using reinforcement learning. As each sample would have its child architecture, doing this will essentially prevent model updates to override past knowledge. This can be illustrated in Figure 1 that the dog sample having a very different probability of building networks in the meta-graph than that of the butterflies, highlighted using a scale of the color panel. As a result, updates that belong to the dog instance will avoid "washing out" the model weights that are very different from itself and mitigated catastrophic forgetting from happening.

## 2  Related Work

**Continual Learning.**   [5] is one of the well-known methods of continual learning. They regularize the parameters that are important for previously learned tasks. [10] fine-tune the last fully connected layers by maintaining a single shared covariance matrix to avoid forgetting. [11] is the work solving continual learning problem with neural architecture search. Although they restrict the overall size of their model, we suspect that it will not be feasible if the incoming tasks are instance-level since the architecture will keep growing which is different from us. [12–14] also belong to the architectural-based category. They isolate the parameter after training old tasks by pruning and embedding to ensure not to forget.

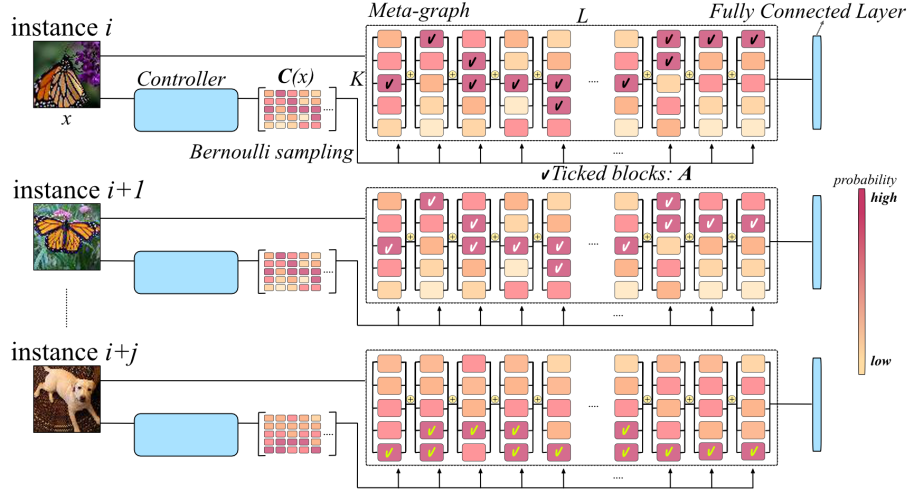

Figure 1: Illustration of the training and inference process. For each input sample $x$, there are three stages: The input sample will first be fed into the controller. Controller will then output an $L \times K$ shape probability matrix $\boldsymbol{C}(x)$ for the sample, where there are $L$ layers and $K$ convolutional blocks per layer. The input sample will then be fed into the child model constructed by meta-graph and sampled architecture $\boldsymbol{A}$, where $\boldsymbol{A} \sim \boldsymbol{C}(x)$ (The ticked blocks in meta-graph, the color of ticks represent $\boldsymbol{A}$ for different instances). For the first two instances that are both butterflies, their sampled architectures $\boldsymbol{A}$ share more similarity than the third instance which is a dog.

The authors of [15] proposed a framework with plenty of experts for each task. The basic idea is similar to us, training several experts for different tasks. However, they have lacked the mechanism for sharing the knowledge between tasks, which may increase the number of parameters and lost the chance for transferring knowledge between tasks. The authors of [16] also proposed a meta-graph-like framework that falls into the architectural-based category. However, the major difference is that they treat each component in the hyper-network as a feature extractor, and during inference time, all trained components will be picked as a single path for all tasks. After training the path for the previous task, the component of the path will be frozen, which may lack the opportunity to re-train the previously trained path when the incoming sample shares the similarity with the old knowledge. Moreover, the major difference between [15, 16] and ours is that they treat their network depends on the task, which might not be suitable for the online continual learning setting. Since the network for each task will overfit the most recent samples without the properly online learning technique, which will face a similar situation that Independent ResNet-18 have to face in Section 4.

**Replay-based** Methods such as [1–4, 17, 18] belong to the replay-based category. The most different premise is that they allow storing a small set of data in a buffer. When training the new task, they will replay the previously seen data from the buffer and make use of it. It may not be a proper premise if the data cannot be store due to the privacy issue, which also contradicts the setting that each sample can only be seen once in the online continual learning setting. Nevertheless, recent works [18] stores learned representation instead of raw data which highly reduces the memory cost and will not violate the privacy issue, which is a promising direction to follow. The advantage of the replay-based method is that they can correct the distribution of the parameter via reviewing the old data to deal with the imbalance logits at the final layer, which is critical for shared output classifier or so-called single-head setting [19, 20]. [19] also shows that it is possible to achieve single-head even with naive rehearsal, so it is possible to apply replay tricks to our method as it is orthogonal to our idea.

**Online Continual Learning.** [2, 1, 17] are the works that study on the scenario that each example can only be used once during training. [4, 3, 10, 18] are the works that also study in a similar setting, however, they also introduce a more challenging shared classifier. Our setting is a combination of these approaches, each sample can only be seen once during training, but we do agree that it is no

need to restrict that the model can only make use of each example once as [3, 4] stated. To our best knowledge, currently, most of the approaches that have a shared classifier required a buffer to store past information [19, 20] or additional data [21] for the learner to review. Since our method is orthogonal to the replay-based approach, we do not introduce a single-head setting in the work.

**Weight Sharing.** According to [22], one-shot architecture search is the approach to reduce computational expenses by using weight-sharing. The weight sharing technique is implemented by training a hyper-network first, then using various searching approaches, for example, reinforcement learning, to find the best architecture for the given training data. We believe that instead of isolating the parameter for different tasks, it is better to solve catastrophic forgetting by weight sharing since it will also benefit from transferring the old knowledge, which shares a similar thought with [6].

**Conditional Computation and Instance Awareness.** The key idea of conditional computation [23] is to make the parameters of the neural network conditioned on the input sample. Several conditional computation methods (e.g., using node sharpening [24], maxout [25], compete to compute [26]) have been proposed to mitigate catastrophic forgetting. Yet, most of these approaches are evaluated on simple feed-forward networks and have not been shown to work on more complex architectures (e.g., convolutional neural networks) or more complex datasets (e.g., CIFAR-10 and CIFAR-100). Instance-awareness is a new concept combining conditional computation and architecture search. InstaNAS [7] leverages the one-shot architecture search consisting of a meta-graph and controller to select a path for each instance (i.e., instance-awareness) in order to improve the overall inference speed and accuracy. However, the concept of instance-awareness has not been explored to mitigate catastrophic forgetting on complex architectures.

## 3 Methodologies

In this section, we describe how we build a model with instance-awareness to solve the catastrophic forgetting problem. The overview is illustrated as Figure 1.

### 3.1 Meta Graph - Controller Framework

In this work, we adopt the framework proposed by InstaNAS [7], which contained two major components: a meta-graph and a controller. The meta-graph (denoted as $G$, parameterized by $\theta$) is a directed acyclic graph (DAG), build with layers of blocks, where each block is a residual block (ResBlocks) from [27] with different kernel sizes, we implement a $L \times K$ matrix as the representation of the meta-graph. We treat every path from source to end as a child architecture $\boldsymbol{A}$, each block can be either picked or not picked. If the controller does not pick any block in a layer, we enforce that it will pass through one of the blocks to ensure the feature map shape is correct. So there will be about $2^{LK}$ possible child architectures for a $L \times K$ shape meta-graph. When passing data, the data is fed into the picked blocks, then we sum the feature maps output from the picked blocks in the same layer and pass it to the next layer. On the other hand, the controller (denoted as $C$, parametrized by $\phi$, received image $x$) is a shallow 3-layer convolutional network that assigns the input image to the downstream path of the meta-graph.

The controller $C$ receives image $x$ as input, then output a $L \times K$ shape of probability matrix $\boldsymbol{C}(x; \phi)$ with $L$ layers and $K$ blocks in each layer. Next, we sample the child architecture $\boldsymbol{A}$, which is also a $L \times K$ matrix, from $\boldsymbol{C}(x; \phi)$ that may be treated as a multinomial distribution for each layer where each single element $\boldsymbol{C}_{l,k}(x; \phi) \in \boldsymbol{C}(x; \phi)$ is sampled by the Bernoulli distribution. Then we construct the child model $m$ with meta-graph $G(\theta)$ and the sampled architecture $\boldsymbol{A}$.

$$\boldsymbol{A} \sim \boldsymbol{C}(x; \phi) \tag{1}$$

$$m(x; \theta_{\boldsymbol{A}}) = G(\boldsymbol{A}; \theta) \tag{2}$$

During inference, after the controller outputs the probability matrix $\boldsymbol{C}(x; \phi)$, we determine the child architecture by $\boldsymbol{A}_{l,k} = \delta(\boldsymbol{C}_{l,k} > \mu)$, where $\delta(\cdot)$ is the indicator function, $\mu$ is the threshold whether picking the block.

To reduce the size of the meta-graph, we adopt the concept from [28]. The concept is also similar to [12, 13], network parameters are often over-parameterized. We merge several kernel options to one single superkernel for each layer to reduce the redundant parameters, where each option will be created during forwarding and backpropagation.

## 3.2 Training the Controller

To keep the meta-graph from saturation, we trained the controller with a multi-objective reward function that maximizes task accuracy meanwhile minimizes the number of blocks picked for the current task. Each transition rewards zero except for the terminal transition, which is calculated as:

$$r(\boldsymbol{A}|(x_i, y_i)) = Acc(\boldsymbol{A}|x_i, y_i) \cdot Sparse(\boldsymbol{A}) \tag{3}$$

$$Acc(\boldsymbol{A}|x_i, y_i) = \begin{cases} 1, & \text{if } y_i = m(x_i; \theta_{\boldsymbol{A}}) \\ 0, & \text{otherwise} \end{cases} \tag{4}$$

$$Sparse(\boldsymbol{A}) = \log(\frac{LK - \sum_{l=1}^{L} \sum_{k=1}^{K} \boldsymbol{A}_{l,k}}{LK}) \tag{5}$$

Where $\boldsymbol{A}$ is calculate by equation 1, $Acc(\boldsymbol{A}|x_i, y_i)$ is to ensure we only calculate reward if the picked child architecture $m(x; \theta_{\boldsymbol{A}})$ classify image $x_i$ correctly, $Sparse(\boldsymbol{A})$ is to measure how much block are used for the instance. The number of blocks is prefer to be as low as possible if all images are classified correctly.

The controller is further trained with policy gradient that is estimated by:

$$\nabla_{\phi} J = \mathbb{E}_{x \sim D, \boldsymbol{A} \sim \boldsymbol{C}(x; \phi)} \left[ (r(\boldsymbol{A}|\cdot) - r(b(x)|\cdot)) \cdot (\nabla_{\phi} \sum_{l=1}^{L} \sum_{k=1}^{K} \log \boldsymbol{C}_{l,k}(x; \phi)) \right] \tag{6}$$

For instance $x$ sampling from dataset $D$, we then sample architecture $\boldsymbol{A}$ from $\boldsymbol{C}(x; \phi)$ to calculate the reward difference between picked architecture $r(\boldsymbol{A}|\cdot)$ and baseline architecture $r(b(x)|\cdot)$, where $b(x) = \delta(\boldsymbol{C}_{l,k}(x; \phi) > \mu)$ is the self-critical baseline that can reduce the variance of instance-wise reward as described in [29].

There exists a potential forgetting occurs in the controller. To deal with this in such a tiny network, we store each controller after training a task which is similar to multi-head. This practical method will not cause overhead since the size of our controller is only $0.2\%$ compare to the meta-graph and it is also better than maintaining a replay buffer. See Section 4 for detailed number and explanation.

## 3.3 Training the Meta-Graph

The training phase of InstAParam contains 2 stages, "pre-train" and "jointly train".

**Pre-train.** One of the key phases of the weight sharing method is to pre-train the *meta-graph*. Since it gives the controller a relatively reasonable environment to search for the child architectures. The way we pre-train our meta-graph is adopted from InstaNAS. First, we train the meta-graph with the data from the first task as the traditional training process (not online). After training half of the epochs, we start to apply "drop-path" in [22], which is to randomly zero out parts of the meta-graph during the pre-train stage.

**Jointly Train.** When training the meta-graph in the "jointly train" phase, we only train the parameters that are picked by the controller in each batch. The loss function of the second task is denoted as $\mathcal{L}_{T2}(\theta_{\boldsymbol{A}}|x)$ which is calculated by the sample-level cross-entropy function for the model $m(x; \theta_{\boldsymbol{A}})$ described at equation 2. Where $\theta_{\boldsymbol{A}}$ denotes the parameter of picked child architecture.

To increase the performance, we do not restrict the blocks that are used in the previous task to be fixed. We allow our model to re-train all blocks, which is similar to the "reused" option in [11]. To furthermore stabilize our training process, we can use either L2 loss or EWC regularization (We used EWC in the experiment). During training, we will only train and regularize the paths picked by the

controller. Following is the loss that we minimized when input samples from the second task arrived after training the first task.

$$\mathcal{L}(\theta|D_{T2}) = \mathbb{E}_{x \sim D_{T2}, \boldsymbol{A} \sim \boldsymbol{C}(x)} \left[ \mathcal{L}_{T2}(\theta_{\boldsymbol{A}}|x) + \lambda \sum_{l=1}^{L} \sum_{k=1}^{K} \delta(\boldsymbol{A}_{l,k} = 1)(\theta_{\boldsymbol{A}_{l,k}} - \theta_{\boldsymbol{A}_{l,k}}^{T1})^2 \right] \quad (7)$$

When sampling image $x$ from task 2 dataset $D_{T2}$, the architecture $\boldsymbol{A}$ will then be sample from $C(x)$ with given image $x$. First, we calculate the loss with parameter $\theta_{\boldsymbol{A}}$ and input sample $x$ as the first term in the square bracket, which is the core idea of our work, train the best child architecture and prevent from interfering with other parameters. Second, we calculate the difference between current parameter at the $k$th block of $l$th layer $\theta_{\boldsymbol{A}_{l,k}}$ and the optimizer parameter of task 1 $\theta_{\boldsymbol{A}_{l,k}}^{T1}$ as a regularization term. The indication function $\delta(\cdot)$ is to restrict that the regularization only applies to the picked blocks. $\lambda$ is a hyperparameter. And similar to EWC [5], we add separate penalties when the third task arrived. One may extend the regularization term of equation 7 with EWC loss.

The "jointly train" phase is to train the controller and the meta-graph by turns. Given an input image, the controller will generate a $L \times K$ matrix $\boldsymbol{C}(x; \phi)$ representing which blocks will be picked to construct the child architecture $\boldsymbol{A}$ for the given image $x$. Then the image will be fed into the model $m(x; \theta_{\boldsymbol{A}})$ constructed by the meta-graph $G$ and the child architecture $\boldsymbol{A}$. After updating the weight of the controller, the weight of the picked blocks in the meta-graph will then be updated.

After training tasks, we do not "pre-train" our meta-graph again, since it might ruin the knowledge without carefully picking suitable paths. As [30] mentioned, the representation similarity between two networks that train with different datasets is higher than the randomly initialize one. Therefore, it is reasonable to keep training the meta-graph with past knowledge. The meta-graph with previous knowledge is a better environment for the controller to search compare to a randomly initialized one.

### 3.4 Encouraging Explorations

Empirically, we find out that the controller will tend to keep finding the child architectures that have been well trained in the previous tasks. Hence, lack of interest to search for other possible architectures. According to the observations and studies [30], the main reason is that instead of picking almost non-trained convolution blocks, the controller still prefers pre-trained blocks even if the visual feature is not close enough. It may also result in suboptimal solutions for new tasks and more importantly overwrites the knowledge we have learned previously, which does contradict the motivation of applying the architectural method, *i.e.* to isolate some information by not overwriting it.

Inspired by count-based exploration in reinforcement learning [31], we encourage the controller to search other blocks when learning new tasks. When jointly training the meta-graph and the controller, we calculate the counted history matrix $\boldsymbol{H}$ which is a $L \times K$ matrix that is element-wise summed by the previously used child architecture $\boldsymbol{A}$ for all instances before the current batch of data. At time $t$, we calculate the counted history matrix $\boldsymbol{H}$ as following:

$$\boldsymbol{H}_t = \sum_{\boldsymbol{A} \in \{\boldsymbol{A}^1, \boldsymbol{A}^2 \ldots \boldsymbol{A}^{t-1}\}} \boldsymbol{A} \quad (8)$$

During "jointly training", we first feed a batch of training data to the controller, it will then output a batch of probability matrix $\boldsymbol{C}(x)$ with size $L \times K$ that represents the probability of picking each convolution blocks. We then subtract the probability matrix $\boldsymbol{C}(x)$ based on the counted history matrix $\boldsymbol{H}$. We regularize the distribution $\boldsymbol{C}(x)$ as follows:

$$\boldsymbol{C}(x)' = \boldsymbol{C}(x) - \gamma \sigma \left( \frac{\boldsymbol{H}^{\frac{1}{2}}}{\max_{l,k} \boldsymbol{H}^{\frac{1}{2}}} \right) \quad (9)$$

The design of $\boldsymbol{H}^{\frac{1}{2}}$ is inspired by count-based exploration. Where $\gamma$ is a hyperparameter, $\max_{l,k} \boldsymbol{H}$ represent the maximum value of matrix $\boldsymbol{H}$ to apply normalizing. $\sigma(\boldsymbol{X}) = \frac{1}{1+e^{-\kappa(\boldsymbol{X}-\epsilon)}}$, which we set $\kappa = 10$ and $\epsilon$ as a matrix filled with value 0.5 in our experiments to shift and curved the sigmoid function in range[0,1], where the input will also be normalized to range[0,1] as equation 9 mentioned.

Sigmoid is to mimic classifying overused blocks. After calculating $C(x)'$, we clamp it into range[0,1] represent the probability of picking each block. With this method, the performance got a noticeable improvement.

# 4 Experiments

We compare Instance-Aware Parameterization (InstAParam) to the baselines in the literature of preventing catastrophic forgetting. We show the benefits of using our approach in terms of finding better architecture for each instance, transferring the knowledge between tasks, and avoiding overwriting the previous information. We compare the performance of our work to the state-of-the-arts in the same online continual learning setting. We also conduct experiments with blurry task boundaries.

## 4.1 Experiment Setup

We conduct the experiments on CIFAR-10, CIFAR-100, and Tiny-ImageNet. We split the dataset into several sets as different tasks. During training, the model receives data one by one in an online learning manner. As [32] mentioned, previous models tend to update the meta-knowledge after learning the entire task, which is not suited for online learning. As mentioned in Section 2, we conduct all the experiments with task descriptors, masks, and controllers (multi-head setting).

The practical way to handle catastrophic forgetting in the controller is storing its parameters after learning each task. The size of the controller is roughly 80K number of parameters, which is fairly small compared to the size of the meta-graph (43M or 15M). On the other hand, to our best knowledge, most reinforcement learning relies on experience replay to handle streaming data, in which the buffer size is often set to 1000 or bigger. When dealing with the CIFAR dataset, which might required $32 \times 32 \times 3 \times 1000 = 3M$ number of parameters, for Tiny-ImageNet, it requires $12M$ number of parameters. Only if there are more than 38 or 150 tasks respectively, our method will have to store more parameters than maintaining a buffer. This also alludes to the advantage of our method, InstAParam does not need more parameters to learn the information if the dataset is more difficult. However, in such severe cases, the buffer size may also need to be enlarged. Moreover, it also contradicts one of the real-world scenarios of online continual learning, not able to store data after seeing it once due to privacy issues.

We set the batch size to 10 for all experiments and train it in an online learning manner, which will discard every batch of received data after updating the weights. We train the models by either SGD or Adam optimizer depends on the default setting of each work and does a simple grid search for how many iterations should it run. Our setting is similar to [2, 3], nevertheless, since we do not focus on the task-free setting in this work, we add a multi-head to GSS-Greedy [3] and RPSnet [16] for fair comparison in all experiments.

We use ResNet-18 [27] as the backbone network of our model in the experiments. We modified the ResNet-18 by expanding the convolution layer in 8 ResBlocks, where each ResBlocks contain 5 options with different kernel and group sizes[1]. To reduce the size of the meta-graph, we replace the convolution layer with superkerenl that provide 4 options of kernel size[2] as InstAParam-single.

## 4.2 Baselines

One Independent Classifier per Task (Independent). One may argue that why not train each task with an independent network. To manifest the ability to find the best architecture for each instance in the online continual learning setting and the power of weight sharing, we set a baseline of training an independent ResNet-18 for each task. Besides, the total number of parameters of this baseline is larger than our model, which can show that although using more parameters, it does not guarantee better performance without properly dealing with the online continual learning setting. Elastic Weight Consolidation (EWC) [5] is to regularize the network parameter during training to prevent parameters move away from the old tasks. Hard Attention to Task (HAT) [13] is the method to isolate the parameters via pruning and masking. Random Path Selection (RPSnet) [16] searches the path for each task randomly and consolidates the knowledge by freezing the previously used path. Averaged

Table 1: Comparison accuracy with state-of-the-art and baseline on CIFAR-10.

| Method | Param. | Accuracy (%) | | | | | |
|---|---|---|---|---|---|---|---|
| | | T1 | T2 | T3 | T4 | T5 | Avg. |
| Independent | 11M × 5 | 85.6 | 70.9 | 78.7 | 86.9 | 85.7 | 81.5 |
| EWC [5] | 11M | 75.2 | 63.9 | 74.2 | 72.0 | 79.9 | 73.0 |
| HAT [13] | 7.1M | 77.1 | 68.5 | 78.6 | 92.2 | 85.1 | 80.3 |
| A-GEM [2] | 11M | 77.4 | 65.2 | 69.2 | 83.7 | 89.8 | 77.1 |
| GSS-Greedy [3] | 11M | 53.6 | 52.1 | 72.3 | 91.7 | 94.3 | 72.8 |
| RPSnet [16] | 80M | 59.2 | 50.1 | 60.9 | 78.5 | 86.4 | 67.0 |
| InstAParam (Ours) | 43M | 89.0 | 73.9 | 75.0 | 90.2 | 90.8 | **83.8** |
| InstAParam-single (Ours) | 15M | 82.0 | 77.1 | 77.7 | 91.1 | 86.8 | 82.9 |

Table 2: The accuracies on split CIFAR-100 and Split Tiny-ImageNet. InstAParam consistently outperform other state-of-the-arts and baseline on a more challenging dataset.

| Method | CIFAR-100 (%) | Tiny-ImageNet (%) |
|---|---|---|
| Independent (pre-train) | 52.3 | 27.2 |
| EWC [5] | 44.0 | 26.6 |
| HAT [13] | 39.2 | 20.3 |
| A-GEM [2] | 50.2 | 31.4 |
| GSS-Greedy [3] | 40.2 | 17.6 |
| RPSnet [16] | 40.1 | |
| InstAParam (Ours) | 54.4 | **36.8** |
| InstAParam-single (Ours) | **55.5** | 33.0 |

Gradient Episodic Memory (A-GEM) [2] modified the origin GEM [1] to speed up the training process. Gradient based sample selection (GSS) is the work that studies on online continual learning, task-free, and blurred task boundary settings. Besides adding a multi-head setting, another tweak for GSS-Greedy is that instead of training with a fraction of the dataset as they stated in their paper, we train all our methods with the full-size dataset.

## 4.3 Quantitative Results

**Experiments on Split CIFAR-10.** To conduct experiments, we divide the dataset into 5 tasks where each task contains two classes. According to our experiment, we found out that there is no need to pre-train on CIFAR-10 before jointly training, so we train all models from scratch. Results are illustrated in Table 1. Here our model has a non-negligible better result than A-GEM [33] and GSS-Greedy [3]. From the results, we also observe that our approach outperforms other methods by a large margin on earlier tasks. This means that our approach can help mitigate catastrophic forgetting especially when the tasks for those that are distant away. For this experiment, the proposed method outperforms state-of-the-arts and the baseline, which shows that InstAParam is a promising method for solving catastrophic forgetting. Most importantly, our model has a non-negligible better result than A-GEM [33] and GSS-Greedy [3] which are methods that focus on online continual learning with the replay mechanism.

**Experiments on Split CIFAR-100 and Split Tiny-ImageNet.** We also conduct experiments on CIFAR-100 and Tiny-ImageNet. We split CIFAR-100 into 10 tasks with 10 classes in each task. Tiny-ImageNet is split into 10 tasks with 20 classes in each task. Different from the previous experiment, we apply the standard pre-train stage as all one-shot architecture search methods will do. We pre-train our meta-graph with the first task, which we know is unfair to compare our model with other state-of-the-arts directly. As a compromise, we pre-train all methods with the first task and exclude calculating the performance of the first task. In Table 2, we see that InstAParam and InstAParam-single outperform all state-of-the-arts in this setting. One interesting observation is that our work outperforms Independent in the setting. We argue that it may because they are not designed to learn from streaming data. Although training independent ResNet for each task may guarantee not to forget on task level, the disadvantage of lacking mechanisms to transfer knowledge and learn each instance

Table 3: Split CIFAR-10 with blurry boundary. The last row Diff. stands for the average accuracy difference with split CIFAR-10 listed in Table 1. InstAParam clearly outperforms other non-replay-based baselines and shows competitive results to replay-based approaches even without buffer.

| Method | Replay | Avg. Acc. | Diff. |
|---|---|---|---|
| EWC [5] | | 62.1 | -10.9 |
| HAT [13] | | 74.6 | -5.7 |
| InstAParam (Ours) | | 83.5 | -0.3 |
| InstAParam-single (Ours) | | 81.8 | -1.1 |
| A-GEM [2] | ✓ | 75.5 | -1.6 |
| GSS-Greedy [3] | ✓ | 82.5 | +9.7 |

is magnified in the setting when each task contains few samples. Surprisingly, InstAParam-single has the best performance for CIFAR-100, even with less number of parameters than InstAParam . It might due to the difference in the number of parameters does not affect performance much in CIFAR-100.

**Experiments on Split CIFAR-10 with Blurry Boundary**    We also test our model in the scenario where the task boundaries are not strictly split, which is similar to the setting in [3]. We split the CIFAR-10 dataset into 5 independent tasks, and randomly exchanged 20% of the data from other tasks. We believe that this might be more challenging for the methods that required a clear boundary between tasks to isolate the parameter or save the information when switching the task [32]. The results are listed in Table 3. Here we divide the methods into two groups: one with replay and one without replay. We see that our method significantly outperforms the baselines approaches without replay. In the table, we can find that even without the replay buffer our model does not suffer much from blurry task boundary setting, which is a problem for methods without replaying mechanisms such as EWC [5] and HAT [13]. As mentioned in Section 2, methods that take advantage of replay usually assume a buffer of samples. As online continual learning only allows samples to be read once, replay-based slightly abuse the setting by allowing it to store samples. They aim to review past knowledge from little data, which is similar to the situation of blurred task boundaries. Nevertheless, our method is on par with the methods that have a replay mechanism. Interestingly, instead of dropping accuracy as other methods do, the accuracy of GSS-Greedy [3] boosts a lot thanks to the tasks that are not disjoint. InstAParam is the only method can deal with blurred boundary even without the mechanism of making use of little data as the replay-based methods do.

## 5    Conclusion

We introduce Instance-Aware Parameterization (InstAParam) to overcome catastrophic forgetting in online continual learning. InstAParam protects the paths in the network according to instance similarity. The mechanism of selecting paths is naturally determined by the controller, which is compact and online updated. The experiment results on CIFAR-10, CIFAR-100, and Tiny-ImageNet show that we have a better ability to handle the problem than previous works. The split CIFAR-10 with blurry boundary experiment shows that InstAParam does not drop the accuracy as severe as other non-replay-based methods. Since NAS is known for its high computational complexity, one straight-line future work for InstAParam is to improve the computational efficiency for enabling even broader applications. Overall, our work presents a novel direction for solving catastrophic forgetting in online continual learning by leveraging the concept of instance-awareness.

## Broader Impact

Our work belongs to the category of continual learning, specifically online continual learning, which is crucial for deep learning model to maintain the knowledge it has learned. It may help deep learning models to keep learning new tasks and integrate the knowledge, eventually become a lifelong learning system. We believe that in the future, AI devices will be a crucial part of society. Our model will be effective when the embedding device required private data to learn and are not able to store the whole data locally.

## Acknowledgement

We are grateful to the National Center for High-performance Computing for computer time and facilities, and Google Research for their support. This research is also supported in part by the Ministry of Science and Technology of Taiwan (MOST 109-2634-F-007-016-), MOST Joint Research Center for AI Technology and All Vista Healthcare.

## Footnotes

[1]specifically [3,1], [5,1], [3,2], [5,2], [3,8].

[2]specifically 1, 3, 5, 7

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
