[Supplementary Material]

# Mitigating Forgetting in Online Continual Learning via Instance-Aware Parameterization (Supplemental)

**Hung-Jen Chen**[1], **An-Chieh Cheng**[1], **Da-Cheng Juan**[2], **Wei Wei**[2], **Min Sun**[134]
[1]National Tsing-Hua University, Hsinchu, Taiwan
[2]Google Research, Mountain View, USA, [3]Appier Inc., Taiwan
[4]MOST Joint Research Center for AI Technology and All Vista Healthcare, Taiwan
andyqmongo@gapp.nthu.edu.tw, accheng.tw@gmail.com
{dacheng,wewei}@google.com, sunmin@ee.nthu.edu.tw

## Appendix A   Symbol Table

Table 1: Symbol Table

| Notation | Meaning |
|---|---|
| $G$ | Meta-Graph, parameterized by $\theta$ |
| $C$ | Controller, parameterized by $\phi$, receive input $x$ |
| $\boldsymbol{C}(x;\phi)$ or $\boldsymbol{C}(x)$ | The probabiltiy matrix, where shape is $L \times K$, output from controller after receiving input $x$. |
| $\boldsymbol{A}$ | An $L \times K$ matrix that represent the sampled child architecture from $\boldsymbol{C}(x;\phi)$ |
| $m$ | Child model, constructed by $\boldsymbol{A}$ and $G$ |
| $L$ | Number of layers for the meta-graph |
| $K$ | Number of modules per layer |
| $\delta()$ | Indication function, value 1 if it is true, value 0 if it is false |
| $\gamma$ | A hyperparameter for count-base exploration |
| $\lambda$ | A hyperparameter for regularizing the weight of the meta-graph |
| $\mu$ | The threshold to decide whether the module in the meta-graph is picked |
| $\theta_{\boldsymbol{A}}$ | Parameters for child architecture $\boldsymbol{A}$ |
| $\theta_{\boldsymbol{A}}^{Ti}$ | Optimal parameters for $\boldsymbol{A}$ on task $i$ |
| $D_{Ti}$ | Dataset for task i |
| $\boldsymbol{H}$ | History matrix, an $L \times K$ matrix to record the usage of each module of the meta-graph. |
| $\sigma()$ | A modified sigmoid function that receive a matrix, range from [0,1], centered at 0.5 |

## Appendix B   Algorithm

**Algorithm 1 :** The training algorithm of InstAParam

**TRAINING**

```
1:    Initialize weight θ of the meta-graph G.
2:    Initialize weight φ of the controller C;
3:    for i in range [1, T]
4:       if i = 1 and pre-train is necessary
5:          Pre-train the meta-graph with D_{T1}
6:       else
7:          for x, y in D_{Ti}
8:             for iteration in range [0,T) do
9:                Get probability matrix C(x) ← C(x; φ)          ▷ Train controller
10:               Encourage controller to search unseen blocks by Eq. 9
11:               Sample child architecture A ∼ C(x)′
12:               Get reward r by Eq. 3
13:               Update φ with policy gradient from r
14:               Update history matrix H ← H + A
15:
16:               Obtain loss L by Eq. 7                          ▷ Train meta-graph
17:               Update θ_A with gradient from L
18:            end for
19:         end for
20:      endif
21:   end for
```

## Appendix C   Ablation Study: Count-Based Search Exploration

We conduct an ablation study to show the strength of count-based search exploration. We compare the performance difference between InstAParam with and without count-based exploration. As we mentioned in Section 3.4, the model will tend to pick the same blocks even after switching tasks, which results in poor performance. Although, InstaNAS tries to solve the problem with "policy shuffling", we found that it does not solve the problem in this scenario. The detailed accuracy is listed in Table 2. As we can see, the forgetting is significant for InstAParam without count-based exploration, the performance drops catastrophically after shifting to the new task.

Besides, we find out that the performance of task 1, $A_{T1}$ boost after training task 4. We suspect that it is due to the similarity between the two tasks. However, it also alludes to a potential problem, the controller tends to pick the architectures that are already well-trained instead of exploring the new blank blocks. Therefore, we adopt the count base exploration idea to encourage our controller to search for other blocks as mentioned in Section 3.4. With the technique of count-based exploration, we successfully avoid the aforementioned problem. Our model reaches almost non-forgetting on split CIFAR-10 and does not sacrifice the initial performance.

## Appendix D   Ablation Study: Weight Regularization

Although we have shown that InstAParam significantly outperforms weight regularization methods such as EWC in Section 4. We would like to further test whether the weight regularization technique we adopt in our model is crucial. We conduct the experiments on split CIFAR-10 with different $\lambda$. As Table 3 listed, even without adopting weight regularization, InstAParam is able to hold relatively high performance compare to one with weight regularization. Although the performance gain is not huge, the main reason we add weight regularization in InstAParam is that we found out it is more stable to train with weight regularization and might also be helpful to achieve higher accuracy.

Table 2: Detailed accuracy of InstAParam for split CIFAR-10. $A_{Tn}$ represent accuracy of task $n$. $M_{Tn}$ stands for model after training task $n$. The model with exploration technique prevent forgetting significantly well compare with the vanilla one.

| | InstAParam w/o Exploration | | | | | InstAParam | | | | |
|---|---|---|---|---|---|---|---|---|---|---|
| | $A_{T1}$ | $A_{T2}$ | $A_{T3}$ | $A_{T4}$ | $A_{T5}$ | $A_{T1}$ | $A_{T2}$ | $A_{T3}$ | $A_{T4}$ | $A_{T5}$ |
| $M_{T1}$ | 91.7 | | | | | 87.1 | | | | |
| $M_{T2}$ | 78.9 | 78.7 | | | | 87.6 | 73.8 | | | |
| $M_{T3}$ | 66.1 | 66.2 | 82.3 | | | 87.4 | 73.7 | 74.9 | | |
| $M_{T4}$ | 34.7 | 52.1 | 63.2 | 92.6 | | 87.3 | 73.9 | 74.5 | 89.9 | |
| $M_{T5}$ | 80.0 | 61.5 | 49.3 | 57.5 | **92.1** | **89.0** | **73.9** | **75.0** | **90.2** | 90.8 |

Table 3: Different $\lambda$. Weight regularization such as EWC slightly helps InstAParam to improve performance.

| EWC $\lambda$ | Accuracy (%) | | | | | |
|---|---|---|---|---|---|---|
| | $T1$ | $T2$ | $T3$ | $T4$ | $T5$ | Avg. |
| 0 | 81.0 | 72.2 | 78.8 | 88.1 | 88.5 | 81.7 |
| 10 | 85.8 | 64.4 | 79.4 | 85.9 | 86.4 | 80.4 |
| 30 | 77.7 | 73.0 | 75.2 | 89.8 | 89.1 | 80.9 |
| 50 | 86.3 | 74.0 | 78.8 | 88.6 | 87.7 | 83.0 |
| 70 | 87.1 | 71.6 | 80.5 | 89.9 | 85.3 | 82.9 |

## Appendix E    Qualitative Analysis: Distribution of Architectures

In this section, we provide a qualitative analysis of how the controller picks child architecture for each instance. First, we will focus on the distribution of the policy for each task.

We visualize the distribution of searched architectures of each task to demonstrate the power of weight-sharing and task awareness using our method. We sum up the probability matrix $C(x)$, for all input samples for each task, and generated the expected architecture distribution $W = \mathbb{E}_{x \sim D} C(x; \phi)$, and color it by the scale in Figure 1. Each element in the matrix represents the probability of picking a certain block in the meta-graph for the task. We find out that each task has its own distribution. The component sharing is only feasible through weight sharing techniques as a result of the meta-graph reuse technique used in the paper. The visualization result is perfectly matched to what we expected when designing the framework, which indicates that our approach achieved a sense of task awareness when dealing with samples. Such a task awareness enables the model to generate different architectures and resulted in updates that do not interfere with each other, which in turn mitigates catastrophic forgetting.

By looking at Fig 1, we may found that it is possible that the controller does not pick any block in the layer. It indicates that the controller does not see the benefit to assign certain blocks. If the controller does not pick any block in a layer, we force it to pass the sample through one of the blocks to maintain the shape of the feature map. The intuition is that if all the block performs the same for the image, it will not cause a problem to assign it to arbitrary blocks. Besides, although the performance is great, there are still plenty of blank blocks in the meta-graph, indicating the room for improvements for further work on space utilization. One direction is to reduce the option in front layers since the differences between those blocks are small.

## Appendix F    Qualitative Analysis: Instance-Awareness

We analyze the instance awareness of our method. We randomly sample some sets of child architectures to demonstrate how the instances with similar visual patterns are assigned to the child architectures with a similar pattern. As we can see in Figure 2, each block represents a group of instances that got assigned to the child architectures that share a similar pattern. As we can see from Figure 2, images sharing similar visual features got assigned to similar neural architectures, which enabled them to update gradients to the architectures that are local to themselves. Such instance-awareness technique is fundamental to mitigate catastrophic forgetting by preventing "washout" past

Figure 1: Expected architecture distribution $W$ for each task, which represents the frequency of each block picked. $W$ is a matrix with shape $8 \times 5$, where there are 8 layers and 5 operators per layer. The color of the matrix is scaled by the value of the element which is in the range[0,1]. We may see that each task has its own distribution, which illustrates the concept each task has its distinct distribution.

Figure 2: Every block represents the images that were assigned to a set of child architectures with similar pattern. As we can see, each block is constructed by images with similar visual pattern.

knowledge that sits in a different architecture that belongs to an image group that is significantly different.

## Appendix G   Implementation Detail

Table 4: Hyperparameter Table for InstAParam

| Dataset | Learning rate of $G$ | Learning rate of $C$ | Iteration per batch | $\gamma$ | $\mu$ | $\lambda$ |
|---|---|---|---|---|---|---|
| CIFAR-10 | 0.0005 | 0.0005 | 10 | 0.25 | 0.5 | 0 |
| CIFAR-100 | 0.003 | 0.03 | 4 | 0.2 | 0.5 | 30 |
| Tiny-ImageNet | 0.007 | 0.03 | 3 | 0.1 | 0.5 | 30 |

We set reward $r(\boldsymbol{A}|x_i, y_i)$ to 30 if child model $m$ classify input $x_i$ correctly, otherwise, set to 0.

For the experiment on CIFAR-100 and Tiny-ImageNet, we first pre-train the model 50 epochs with SGD optimizer with a learning rate 0.01.

The meta-graph is trained by the SGD optimizer and the controller is trained by Adam optimizer.

For dataset CIFAR-10 and CIFAR-100, we apply simple data augmentation such as normalize, flip, and crop. For Tiny-ImageNet, we add cut out data augmentation.

Note that for easier implementation, we implement InstAParam-single by duplicating part of superk-erenl to create dummy weights during forwarding and backwarding as we mentioned in the main paper. One may re-implement the mechanism by reusing the existing kernel.

Table 5: Dataset splits

| Dataset | CIFAR-10 | CIFAR-100 | Tiny-ImageNet |
|---|---|---|---|
| Num of tasks | 5 | 10 | 10 |
| Num of classes per task | 2 | 10 | 20 |
| Num of samples per train-set | 10000 | 5000 | 10000 |
| Num of samples per test-set | 2000 | 1000 | 1000 |

## Appendix H    Detailed Accuracy for InstAParam-Single

Figure 3: CIFAR-10. We record the accuracy of InstAParam-Single during training on split CIFAR-10. We can see that our method will not forget catastrophically while switching to the future task.

Figure 4: CIFAR-100. We record the accuracy of InstAParam-Single during training on split CIFAR-100. Note that we pre-train the meta-graph with task 0.

Figure 5: Tiny-ImageNet. We record the accuracy of InstAParam-Single during training on split Tiny-ImageNet. Note that we pre-train the meta-graph with task 0.