[Reviews · NeurIPS 2020]

Review 1

Summary and Contributions: The authors propose a method for overcoming forgetting even in online settings using recent ideas from architecture search. They utilized a controller module that is trained in a pretraining stage that then selects paths through the network.

Strengths: - Interesting combination of ideas - Experimental results on challenging datasets are promising.

Weaknesses: A few things that might be weaknesses or need to be clarified - Computational complexity of the updates, in the online setting this is critical, it should be discussed and compared to the other methods. - Pretraining phase could be described more clearly. Currently it is only mentioned in L279-285. It is my understanding that an offline pretraining is done on the controller but using only the first task, does the controller then adapt online for subsequent tasks? -Experiments: Why do the authors not compare to ER[1], why are teh results shown weaker for example for CIFAR100 on A-GEM than those reported in [1]. It would be good to also report performance after each task and as well forgetting metric - (minor) The authors note the memory o the controller storage can be large, although the reviewer is not too concerned of this. Do the authors have thoughts about how to adapt this to the shared-head / class incremental settings?

Correctness: Yes

Clarity: The paper is overall well written. (minor) I suggest the authors make it more clear in Fig 1 those are convolutional (or generally multi parameter) blocks. The figure and position of the FC layer text make it seem initiailly like those are MLP layers.

Relation to Prior Work: I found the literature review thorough.

Reproducibility: Yes

Additional Feedback: Overall I found the paper and direction promising and an interesting insight about the connection of recent architectures search methods, continual learning, and conditional computation. I have a few potential concerns mentioned above but will consider further increasing my score.


Review 2

Summary and Contributions: This paper studies online continual learning in a multi-headed setting, meaning that the agent knows the task it is supposed to predict at test time. An input is fed into a controller that determines what parts of the network should be used for training.

Strengths: While sparse updates and modules have been discussed in the community for mitigating catastrophic forgetting, their approach is interesting and novel.

Weaknesses: The justification for not studying the "single head setting" is weak. The multi-headed setting where task labels are used during testing for inference makes the task much easier: https://arxiv.org/abs/1910.02509 (see Table 4) https://arxiv.org/pdf/1801.10112.pdf (see Table 1) They ignore other recent work on continual online learning from 2019: https://arxiv.org/abs/1809.05922 https://arxiv.org/abs/1909.01520 https://arxiv.org/abs/1910.02509 A comparison of the number of additional parameters/storage needed for each method is needed. The authors argue against replay methods, but the replay buffer size needed for those methods is likely much smaller in terms of memory than the 43M+ additional parameters they add in. I'm not sure if the multi-headed paradigm has much value for studying today. A deployed agent would rarely have access to the necessary information to select the output head and it isn't needed by many recent methods: iCaRL - CVPR-2017 - https://arxiv.org/abs/1611.07725 EEIL - ECCV-2018 - https://arxiv.org/abs/1807.09536 Unified Classifier - CVPR-2019 - http://openaccess.thecvf.com/content_CVPR_2019/papers/Hou_Learning_a_Unified_Classifier_Incrementally_via_Rebalancing_CVPR_2019_paper.pdf BiC - CVPR-2019 - https://arxiv.org/abs/1905.13260 IL2M - ICCV-2019 - http://openaccess.thecvf.com/content_ICCV_2019/html/Belouadah_IL2M_Class_Incremental_Learning_With_Dual_Memory_ICCV_2019_paper.html ScaIL - WACV-2020 - https://arxiv.org/abs/2001.05755 Deep SLDA - arXiv-2020 - https://arxiv.org/abs/1909.01520 REMIND - arXiv-2020 - https://arxiv.org/abs/1910.02509 The experimental setup is confusing and I am not exactly sure how they are training the controller. They are storing data, so I don't know if that makes it any better than a replay buffer.

Correctness: I think so, but see weaknesses.

Clarity: The paper has many grammar problems. I am confused about the experimental setup.

Relation to Prior Work: No -- the discussion of the prior work on continual learning methods needs work.

Reproducibility: Yes

Additional Feedback:


Review 3

Summary and Contributions: The paper proposes to use dynamic architectures (InstaNAS [7] specifically) along with tricks from other related work for task-incremental online learning (no revisiting samples, task identity known for all samples). Similar looking samples are (theoretically) routed similarly, while dissimilar samples use different parts of the network to help mitigate catastrophic forgetting. The paper shows 4-5% performance boosts for CIFAR10, CIFAR100, and TinyImageNet.

Strengths: - Motivation Despite similar approaches in the literature, using one-shot NAS methods (which are relatively new) for incremental learning is relatively new and a good idea. Routing based on instance similarity is neat (although also present in related work). - claims The experiments are relatively well-designed, taking into account of fairness well most of the time. Method design is reasonable without too many ad-hoc components that are under-explored. - relevance The problem, task-incremental online learning, is quite relevant to NeurIPS despite the relatively narrow scope (the intersection of task-IL and online learning).

Weaknesses: - significance and novelty: is quite similar to closely related work [13,14] (and maybe PathNet), but does not compare to them. -- The discussion says "the major difference between [13,14] and ours is that they treat their network depends on the task, which might not be suitable for online continual learning setting." So the only difference is that they may underperform. This paper either has to compare to [13,14] (especially [13] is very similar and open-source), or show numbers in efficiency (FLOPs, test time, etc) to show that they cannot be used efficiently. - claims and evaluation -- fairness of number of parameters: As far as I can tell, HAT has 7.1M params, A-GEM uses ResNet18 which is 11M params. This paper uses 43M params which is not fair. It is also not fair to say all methods have similar efficiency at test time, since multiple blocks in the same layer can be on. -- The exploration trick improved performance quite a lot. It is great that this paper did an ablation study, but since it has a lot of hyperparameters and design choices (why H^1/2 not H, why sigmoid, how are gamma/kappa/epsilon chosen), it is unclear how the design choices are made and how the hyperparameter was tuned, as this would tweak the balance between learning and remembering. An analysis of sensitivity to hyperparameters is preferred. -- The paper is not *entirely* an online method, since the model is pre-trained on the first task (with multiple epochs) to make NAS training work. --- Was line 253 epochs referring to this? Or did this paper train anything else with more than 1 epoch?

Correctness: The claims, method, and empirical analysis is mostly correct (see weaknesses).

Clarity: There are several spots of vagueness: - How the blocks in the meta-graph are connected. Do they use different connection layers between blocks in neighboring layers? How is ResNet18 split in blocks? - Did this paper end up using L2 or EWC loss for the weights? - Line 218 contradicts with eq. 8 -- was A used or was C used?

Relation to Prior Work: Yes, but the reason for not comparing is lacking.

Reproducibility: No

Additional Feedback: The idea is fine and the results are nice, but the experiments are not very convincing for me to say that the proposed method outperforms prior work or closely related work on a fair ground. Please address the weakness section (everything except the "not entirely online" comment) with arguments or additional experiments. -------------------------------------- Update: The rebuttal adequately addresses most of my concerns except for hyperparameter sensitivity. Although please double check that your score increases 54.5->55.5 by decreasing number of parameters. Also please include the details (e.g. about how to do single-Path NAS) and address requested clarifications. I'm increasing the score.


Review 4

Summary and Contributions: They adopted the concept of "instance awareness" in the neural network to alleviate the catastrophic forgetting problem. They proposed a method to protect the path by restrticting the gradient updates of one instance from overriding past updates calculated from previous instances if these instances are not similar. Finally, they achieved the best results in CIFAR-10, CIFAR-100, and Tiny-ImageNet tests.

Strengths: The algorithm is novel. Also, they did not use the replay buffers, but it shows competitive results.

Weaknesses: The proposed method is good, but the experimental results are not convincing because they just compared the proposed algorithm with only four methods such as EWC, HAT, A-GEM, GSS-Greedy. They need to compare it to other state-of-the-art algorithms.

Correctness: Yes.

Clarity: It is easy to follow.

Relation to Prior Work: Yes.

Reproducibility: Yes

Additional Feedback: I will keep my score after reading the rebuttal.

[Author Response · NeurIPS 2020]

We thank you for the valuable reviews and the agreements on the novelty of our
method. We will fix the clarity issues and grammar problems in the final version.

| Method | Param. | Accuracy (%) | |
| --- | --- | --- | --- |
| | | CIFAR10 | CIFAR100 |
| Ours | 43M | 83.8 | 54.4 |
| Ours | 15M | 81.8 | 55.5 |
| [14] | 80M | 67.4 | 40.1 |

**Reviewer 1**  Thanks for the positive review, which encouraged us a lot.
*Confuse about the pretrain stage.*  We have to clarify that the pretrain stage is to
train the meta-graph instead of the controller in offline learning with the initial task.
Then we jointly train both controller and meta-graph in an online learning manner.
We will make it clear in the final version.
*Computational complexity of update ... in online learning.*  The computational
complexity of our method is high due to the NAS-based model.  However, our
method "improve the accuracy" consistently in the scenario when the models are not able to store or see each mini-batch
of samples after doing several times of backward propagations, i.e. instance-level forgetting problem.  Improving
computational complexity is a practical direction to study in the future.
*Why not compare to [1] and the performance difference.*  We compare to A-GEM instead of [1] since A-GEM shows
better performance and is architecturally similar to [1].  The number of classes per task in our work is also used by [14,
17], but different from A-GEM.  Hence, we cannot compare with the number in A-GEM directly.
*Storage of controller could be large.*  We clarify that our controller only requires 80K number of parameters, which is
highly efficient comparing to storing the raw data.
*Potential for adapting to Single-head.*  Most methods use the replay mechanism to achieve single-head.  Note that
labeled examples are used for dealing with the imbalance logits at the last layer.  Hsu et al.[1]  shows that it is possible to
achieve single-head even with naive rehearsal, so it is possible to apply replay tricks as it is orthogonal to our idea.

**Reviewer 2**  Thanks for agreeing that our idea is novel and providing more related works including brand new papers
accepted by ECCV'20 and CVPR-W'20. We will cite those in our final version.
*Value and difficulty of multi-head.*  We agree that single-headed is interesting and more challenging. However, online
continual learning with multi-headed is still an unsolved problem, where our method achieves consistent improvements
over several methods on different datasets. Moreover, our method can be adapted to single-head by applying the replay
mechanism since replay is orthogonal to our proposed idea.
*Confuse with experiment setup and why it is better than replay buffer.*  We store the learned parameters (80K) of the
controller for each task. This is more efficient than replay buffer which increases proportionally to the size of the dataset.
Directly storing data in the buffer also violates our setting of not able to see past examples, which might be restricted
when security is a concern.
*Comparison of number of parameters.*  We have compared the additional parameters in line 241 of our main paper.
Our method only required 80K number of additional parameters per task compare to the standard replay-based method,
which required 3M for CIFAR, 12M for Tiny-ImageNet. Besides, the size of our model (meta-graph) can be shrunk from
43M to 15M by adopting Single-Path NAS[2] (see the performance in the table above), while other architectural-based
methods [14] required 80M, [13] required 66∼74M.

**Reviewer 3**  Thanks for appreciating the idea, we do several experiments and arguments to make it more convincing.
*Number of parameters.*  Please refer to the table above and the response to R2, we have shrunk our model size from
43M to 15M by adopting Single-Path NAS. Another architectural-based method [14] has more parameters compare to
us. The number of parameters for autoencoders and expert networks in [13] will increase with the number of tasks, so it
is not as efficient as ours.
*Compare to [13,14].*  We report the performance of [14] in the table above. Worse performance is expected as explained
in line 96-97 of the main paper. Note that we do not show the performance of [13] as we failed to achieve reasonable
performance with the authors' MatLab code in our setting. However, the worse result is also as expected since the
architectural-based methods often fail to handle online learning settings without proper design and tuning.
*Design choices of count-based exploration.*  The design of $H^{\frac{1}{2}}$ is inspired by count-based exploration, sigmoid is to
mimic classifying overused blocks. $\epsilon$ is set to 0.5 since the input was ranged in [0,1]. Other hyperparameters analysis
will be included in the final paper.
*Not entirely Online Learning and confused about "epochs".*  For the concern of not entirely online learning, please
refer to the answer to R1's 1st and 2nd questions. For pretraining, we train the meta-graph on the whole task with
multiple epochs. For jointly training, we will back prop the model with received mini-batch several times and not seeing
it anymore. We would like to clarify that the term "epoch" refers to iterating through a mini-batch instead of the whole
task, we will modify it as "iteration" in the final version.

**Reviewer 4**  Thanks for appreciating our work, we compare with RPSNet [14] in the table above.

## Footnotes

[1]Hsu, Yen-Chang, et al. "Re-evaluating continual learning scenarios: A categorization and case for strong baselines." arXiv preprint arXiv:1810.12488 (2018).

[2]Stamoulis, Dimitrios, et al. "Single-path nas: Designing hardware-efficient convnets in less than 4 hours." Joint European Conference on Machine Learning and Knowledge Discovery in Databases. Springer, Cham, 2019.


[Meta-Review · NeurIPS 2020]

The initial evaluations of this paper were somewhat mixed. However, the response provided by the authors answered many of the questions the reviews had raised including ones regarding computation cost and the number of additional required parameters. I find that this work opens up new research avenues and should be published. I urge the authors to take the reviewers' comments into account in preparing their camera-ready version. In particular, I would suggest: 1) Adding a discussion regarding the computational complexity of the method (even if improvements are left to future work), 2) Mentioning and contrasting to 2019/2020 work as suggested by reviewer #2 (papers in ICRA'19, ECCV'20 to appear, and CVPR'20 workshop), 3) Including the new results (e.g., table at the top of your response), 4) Add a discussion of the number of required parameters especially as it compares to other methods (this was clearly explained in your author response). There is still one reviewer who feels relatively strongly that this manuscript is not yet ready to be published. My understanding is that the reviewer's main argument is that the field of continual learning has started to and should move away from multi-head setups. While I agree that practical settings are more likely to require single-head settings and many recent methods study the single-head setting, I also do not think that it would be fair for the conference to reject this paper solely based on this (e.g., papers with multi-head setups continue to be published elsewhere). Further, as pointed out by a reviewer in the discussion and by the authors in their response, many multi-head approaches can be adapted to the single-head setting and so the proposed approach is orthogonal to the question of single-head vs. multi-head. It may be good to say this explicitly in the paper.